# The Effects of Green Tea Catechins in Hematological Malignancies

**DOI:** 10.3390/ph16071021

**Published:** 2023-07-18

**Authors:** Fernanda Isabel Della Via, Marisa Claudia Alvarez, Rosanna Tarkany Basting, Sara Teresinha Olalla Saad

**Affiliations:** Hematology and Transfusion Medicine Center, University of Campinas/Hemocentro, UNICAMP, Rua Carlos Chagas 480, Campinas 13083-878, SP, Brazil; fer_idvf@hotmail.com (F.I.D.V.); rosannatb@gmail.com (R.T.B.); sara@unicamp.br (S.T.O.S.)

**Keywords:** epicatechin, epigallocatechin, epicatechin-3-gallate, epigallocatechin-3-gallate, catechins, leukemia

## Abstract

Green tea catechins are bioactive polyphenol compounds which have attracted significant attention for their diverse biological activities and potential health benefits. Notably, epigallocatechin-3-gallate (EGCG) has emerged as a potent apoptosis inducer through mechanisms involving caspase activation, modulation of Bcl-2 family proteins, disruption of survival signaling pathways and by regulating the redox balance, inducing oxidative stress. Furthermore, emerging evidence suggests that green tea catechins can modulate epigenetic alterations, including DNA methylation and histone modifications. In addition to their apoptotic actions, ROS signaling effects and reversal of epigenetic alterations, green tea catechins have shown promising results in promoting the differentiation of leukemia cells. This review highlights the comprehensive actions of green tea catechins and provides valuable insights from clinical trials investigating the therapeutic potential of green tea catechins in leukemia treatment. Understanding these multifaceted mechanisms and the outcomes of clinical trials may pave the way for the development of innovative strategies and the integration of green tea catechins into clinical practice for improving leukemia patient outcomes.

## 1. Green Tea

Green tea is one of the oldest and most consumed beverages [1,2,3,4,5]. Originating in China and Southeast Asia [5], the beverage is made from the dried leaves of the *Camellia sinensis* plant. Green tea has a variety of reported health benefits [3], such as anticarcinogenic, anti-diabetic, anti-inflammatory, anti-microbial, anti-obese, antioxidant and cardioprotective properties [4,6].

The production of green tea involves the pan-frying or steaming of the freshly harvested leaves [2,3,5] in order to inactivate the polyphenol oxidase and other enzymes that preclude the fermentation/oxidation, preserving the active chemical characteristics [2,3,5,7]. The chemical composition of green tea comprises a diverse range of compounds, such as alkaloids (caffeine, theobromine and xanthine), amino acids (arginine, aspartic acid, glutamic acid, glycine, leucine, L-theanine, lysine, serine, threonine 5-N-ethylglutamine, tryptophan, tyrosine and valine), carbohydrates (cellulose, glucose and sucrose), catechins (polyphenolic compounds), fiber, lipids (linoleic acid and α-linolenic acid), pigments (carotenoids and chlorophyll), trace elements (calcium, chromium manganese, copper, iron, magnesium, nickel, selenium, sodium cobalt and zinc) and vitamins (A, B2, B3, C, E and K) [3]. The composition varies according to environmental circumstances (such as the soil, climate, light, geography, microorganisms and temperature), horticultural practices, variety and age of the leaves [1,5].

Catechins are the major bioactive compounds present in up to 30% of the dry green tea leaf weights [2,3,4] and are probably responsible for the cancer-preventive effects observed [8]. Catechin fractions include epicatechin (EC), epigallocatechin (EGC), epicatechin-3-gallate (ECG) and epigallocatechin-3-gallate (EGCG) [2,3,4] (Figure 1). EGCG is the major bioactive polyphenol in green tea, either as the most abundant compound, accounting for 50-75% of the total quantity of catechins, or as the most active and potent catechin [2,3,5,9,10,11].

## 2. Induction of Cell Death in Acute Leukemias by Green Tea Catechins

The induction of programmed cell death or apoptosis plays an important role in the treatment of cancer as it is a target strategy for the elimination of cancer cells [12,13,14]. All green tea catechins (or polyphenols) possess this feature and in different types of cancers, either in vitro or in vivo, for example in myeloidasa [15,16,17,18,19,20,21,22,23,24,25,26,27,28,29,30,31,32,33,34,35] and lymphoid leukemia [11,19,30,36,37,38,39,40,41,42,43,44,45,46,47,48,49], and this action is dose- and/or time-dependent [15,17,19,20,21,22,24,25,26,27,29,33,34,36,37,38,39,41,45,49,50,51] (Table 1).

Morphological change hallmarks of apoptosis, such as nucleus and chromatin condensation, nuclear fragmentation and apoptosis bodies, were also observed with the use of green tea polyphenols [11,20,22,23,26,29,30,33,34,35,36,37,40,41,44,46,47,48,52]. In fact, green tea catechins can induce apoptosis via intrinsic pathways through the modification of the redox system/oxidative stress. The elevation of reactive oxygen species (ROS) levels by catechins [26,27,33,45] induces the loss of the mitochondrial transmembrane potential (∆Ψₘ), with a subsequent release of cytochrome c from the mitochondria into the cytoplasm and caspase activation [11,12,13,20,21,26,33,43,45,47,53]. Moreover, catechins further alter the expression of BAX, BCL-2, MC-1, survivin, XIAP [12,14,20,26,42,43,45]. Interestingly, some authors reported that in the presence of antioxidants such as N-acetyl-L-cysteine (NAC) [26], catalase or manganese superoxide dismutase (Mn-SOD) [27,45], the effect of catechins is lower or blocked; however, the addition of antioxidant system inhibitors, such as buthionine sulfonimines (BSOs), enhanced apoptosis [53]. Moreover, green tea polyphenols can induce the extrinsic pathway, since EGCG can bind to the cell surface of FAS [23,39], activating caspase-8 [23,35].

Saeki et al. (2000) demonstrated that green tea catechins with a pyrogallol-type structure in a B-ring, or in 3-*O*-ester, contribute to apoptosis. They further suggested that a 3-*O*-gallate group in a cis-relationship with the B-ring accelerates the activity [49].

Lee at al. demonstrated that EGCG is a tyrosine kinase inhibitor that halted vascular endothelial growth factor (VEFG) signaling in chronic lymphoid leukemia B (CLL B) cells by inhibiting R1 and R2 receptors (based on reduced levels of phosphorylation) and decreasing p-STAT3 levels, resulting in caspase activation and downregulation of Mcl-1 and XIAP (dose-dependent) [42,43].

The induction of apoptosis observed by Britschgi et al. is aligned with increased death-associated protein kinase 2 (DAPK2) at mRNA and protein levels. They also observed that the toxicity of EGCG correlates with the expression of 67 kDa laminin receptor (67LR) in primary AML blasts [16]. Supporting these findings, Asano et al. (1997) described a dose-dependent decrease in the proliferation rate of blast cells from AML patients induced by EGCG (acute myeloblastic leukemia) [15]. The authors proposed that EGCG causes growth inhibition by non-specific competition between EGCG and receptors of hematopoietic growth factors—an “EGCG sealing effect”—as reported in studies showing that EGCG inhibited the modulation of c-Kit, a receptor for stem cell factor on leukemic cells [15].

In mice xenograft models, green tea or EGCG reduced tumor weight and inhibited tumor growth [26,54] by reducing the number of mitoses [26], induction of apoptosis, cell cycle arrest and autophagy [54]. Interestingly, the mice showed no side effects [26]. In an acute promyelocytic leukemia (APL) model (PML/RARα), green tea treatment reduced blast percentages (CD34 and/or CD117) in the bone marrow and spleen possibly through the induction of apoptosis and reduction in the CXCR4/HIF-1α signaling pathway in response to a decrease in ROS levels [55]. In addition, EGCG treatment of this mouse model modulates the Bax, Bad, Bcl-2 and c-Myc, parallel to the induction of differentiation [7]

The induction of apoptosis by green tea extract may additionally be associated with the inhibition of proteasomal activity [11,38,56] leading to increased levels of ubiquitinated proteins [38]. EGCG is a potent and specific inhibitor of proteasomal chymotrypsin-like activity [56].

**Table 1 pharmaceuticals-16-01021-t001:** Effects of green tea catechins in apoptosis and cell growth.

Model	Compound/Component	Effect	Reference
**In vitro**			
MOLT 4B	C, EC, ECG, EGC and EGCG 0.025–0.100 mM (3 days)	Inhibited the growth and apoptosis detected by DNA fragmentation and morphological change and inhibited the ornithine decarboxylase (ODC) activity (↓)	[36]
Leukemia blast cells from AML patients HEL	EGCG 0–100 nM (8–72 h)	Inhibited proliferation and the effects of TNFα or TPA (↓) and down modulated c-Kit (↓)	[15]
HL60	Tea polyphenols 60–4000 µg/mL (5–48 h)	Induced apoptosis detected by internucleosomal DNA degradation, DNA ladder and apoptotic vehicles	[34]
U937	EGCG 100–400 µM (16 h)	Induced apoptosis detected by formation of DNA ladder, chromatin condensation and apoptotic bodies	[48]
U937	EGCG 100–400 µM (6–16 h)	Induced apoptosis (DNA fragmentation)	[49]
U937 JURKAT	EGCG IC50 = 26.0 µM (12 h) IC50 = 25.3 µM (12 h)	Inhibited cell growth (↓ ATP content), induction of apoptosis (morphological changes, chromosome condensation and DNA fragmentation ladder)	[46]
Peripheral blood T-lymphocytes of acute T-cell leukemia patient KODV	Tea (green tea polyphenols) EGCG 3-27 µg/mL (3 days)	Inhibited cell growth, induction of apoptosis (DNA fragmentation) and suppression of HTLV-1 pX mRNA	[44]
HL60 JURKAT K562	EGC 50 µg/mL (2–24 h)	Induced apoptosis (DNA fragmentation and PARP cleavage)	[30]
U937	EGCG 200 and 400 µM (0–16 h)	Induced apoptosis (formation of apoptotic bodies, DNA ladder formation, ↑ caspase 8 activity and interaction with Fas)	[23]
U937	EGCG and GHF (high molecular weight fraction from green tea) EGCG—200 µM IC50 = 49 µM GHF 1.2 mg/mL IC50 = 0.61 mg/mL (20 h)	Induced apoptosis (formation of apoptotic bodies and DNA ladder formation)	[22]
HL60	EGCG	Induced apoptosis (internucleosomal DNA fragmentation) and decreased activities of MnSOD and CuZnSOD	[32]
L1210	EGCG 200 µM (24 h)	Induced apoptosis (DNA ladder)	[52]
NALM6 Health human lymphocytes	EGCG 10–100 µM (24 h)	Reduced viability by DNA damage and PAR formation (poly (APD-ribosyl) (↑))	[37]
K562 V-79	EGCG 50–500 µM (1–48 h)	Inhibited cell growth (↓ thymidine incorporation) and induction of apoptosis (chromatin condensation, nuclear and DNA fragmentation and ↑ caspase 3 and 8 activity) Protection of normal cells from genotoxic or carcinogenic assault	[47]
WEHI-3B JCS	EGCG 0–40 µM (48 h) IC50 *=* 16.8–31.0 µM	Inhibited proliferation (↓ thymidine incorporation), the ability to form colonies and induce apoptosis (formation of DNA ladder, condensed and fragmented nuclear structure)	[50]
JURKAT	EGCG and synthetic analogs of EGCG (with modification in the A-ring, C-ring or ester bond) 0–2.5 µM (0–24 h)	Induced apoptosis by inhibiting proteasomal activity (DNA fragmentation) Induced cycle arrest (↑ G1 population, ↑ sub-G1 DNA cell population)	[56]
JURKAT NIH/3T3	EGCG and green tea extract 0–50 µM (0–48 h)	Induced apoptosis by inhibiting proteasomal chymotrypsin-like activity (↑ ubiquitinated proteins, PARP cleavage and caspase-3/-7 activation)	[38]
Primary chronic lymphocytic leukemia B-cells Human splenic B-cells	EGCG 0–40 µg/mL (24 h)	Induced apoptosis (caspase 3 activation, PARP cleavage); suppression of Bcl-2, XIAP and Mcl-1 (↓) and VEGF-R1 and VEGF-R2 phosphorylation (↓)	[43]
Primary chronic lymphocytic leukemia B-cells Human splenic B-cells	EGCG 3.12–25 µg/mL (24 h)	Induced apoptosis, inhibition of VEGF receptor activation, ↓ levels of serine p-STAT3 and ↓ Mcl-1 and XIAP	[42]
HL60 K652	Green tea (GT)/EGCG 0–1000 µg/mL (24 h) IC50 HL60 GT = 375/EGCG = 60 K562 GT = 400/EGCG = 58	Induced apoptosis (chromatin condensation, nuclear fragmentation, apoptotic bodies, cell shrinkage and ↓ thymidine incorporation in nuclear DNA)	[25]
HL60 K562	EGCG 0–100 µg/mL (24 h)	Induced apoptosis (chromatin condensation, nuclear fragmentation, DNA fragmentation, caspase-3/-8 activation, ↓ Bcl-2 ↑ Bax)	[35]
UF-1 NB4 Fresh cells from patients with AML	EGCG 100 µM (0–24 h) IC50 = 50 µM	ROS production, induced apoptosis: mitochondrial dysfunction; release of cytocromo C; Bax ↑, Blc-2 and survivin ↓; caspase-3 ↑; PARP cleavage; DNA ladder. Induced cycle arrest—↑ G1 phase and ↓ S phase; ↑ expression of p21 and p27	[26]
IM9, RPMI8226 and U266 HS-sultan Bone marrow samples from multiple myeloma	EC, ECG, EGC and EGCG 0–100 µmol/L (0–72 h) IC50 (EGCG) HS-sultan 17 µmol/L IM9 20 µmol/L	Induced apoptosis through ROS production (↑): mitochondrial dysfunction—loss of Δψm (↓); release of cytocromo C, Smac/DIABLO and AIF; Bax ↑, Blc-2 and Mcl-1↓; caspase-3 and -9 ↑; morphologic changes—condensed chromatin, nuclei fragmented and apoptotic bodies; and DNA ladder EGCG (10 µmol/L) + AS_2_O_3_ (2 µmol/L) intensified apoptosis and the production of ROS Induced cycle arrest—↑ G1 phase and ↓ S phase	[45]
U937	EGCG 100 µM (8–24 h) GHF (high molecular weight fraction of green tea) 0.6 mg/mL (8–24 h)	Induced apoptosis (DNA fragmentation) GHF also induced cycle arrest (↑ G2/M, ↑ p21/Waf1 (mRNA and protein expression)	[28]
K562	Green tea extract 100 mg/mL (0–24 h)	Induced apoptosis (chromatin condensation, nuclear fragmentation, DNA fragmentation, ↓ Bcl-2, ↑Bax and caspase-3/-8 activation) Induction of cycle arrest (↑ sub G1 peak, ↓DNA content of G1 phase)	[17]
K562 U937 Primary leukemic cells—CML and ALL (Ph+) Normal WBC	TRE—tea (*Camellia sinensis* var *assamica*) root extract 0–15 µg/mL	Induced apoptosis (↓ the rate of cellular DNA synthesis—↓ thymidine incorporation, DNA degradation, apoptotic bodies and membrane blebbing) Induced cell cycle arrest (↑ content of hypoploid DNA and ↓ content DNA in G0/G1 phases—U937 or ↓ cells in S or G2/M phases—K562)	[19]
RAJI	EGCG and synthetic analogs of EGCG (a para-amino group on the D-ring) 25 µM (4–24 h)	Induced apoptosis (caspase-3 activation, PARP cleavage) and inhibition of proteasome activity (accumulation of proteasome target protein, like Bax, Iκb-α and p27)	[11]
MPO-positive myeloid leukemia cell lines: HL60, KASUMI, NB4 and UF-1 MPO-negative myeloid leukemia cell lines: KG1, K652, THP-1 and U937	EGCG 0–300 µM (30 min–48 h)	Induced apoptosis in MPO-sensitive leukemia cells through ROS production	[27]
HL60 K562	EGCG and 5-AZA-CdR	Induced apoptosis through mitochondrial dysfunction—loss of Δψm (↓) and modulation of Bcl-xl (↓) and BAX	[57]
HTLV-I-positive ATL cell line: C91-PL HuT-102 HTLV-I-negative cell line: CEM JURKAT	EGCG 0-400 µM (0–96 h) IC50 (48 h) C91-PL= 310 µM HuT-102 = 350 µM CEM = 272 µM Jurkat = 378 µM	Induced apoptosis (DNA fragmentation, ↑ pre-G1 phase cells, ↑ p21, p53 and Bax, ↓ Bcl-2α, ↓ TGF-α—cytokine with proliferative activities, ↑ TGF-β2—anti-proliferative and apoptotic effects; DNA fragmentation)	[40]
HL60 undifferentiated	EGCG 50 µM (4 h)	Induced apoptosis detected through the formation of apoptotic bodies and DNA ladder	[29]
HL60 V79-4	GTP 0–300 µg/mL (0–72 h) IC50 = 49.5 µg/mL (48 h) IC50 = 50.0 µg/mL (72 h)	Induction of apoptosis detected by nuclear fragmentation; Bcl-2 ↓; PARP cleavage (↑) and pro-caspase-3 ↓ Induced cycle arrest—↑ sub-G1 phase No cytotoxic effects in V79-4 with GTP (48 h)	[21]
HL60 V79-4	EGCG and EGC 50 µM (24–48 h) EGCG IC50 = 60 µM (48 h) IC50 = 57.7 µM (72 h) EGC IC50 =107.7 µM (48 h) IC50 = 97.5 µM (72 h)	Induced apoptosis detected through nuclear fragmentation; Bcl-2; and pro-caspase-3 ↓ EGCG > EGC No cytotoxic effects in V79-4 with EGCG or EGC (48 h)	[20]
Primary AML blasts cells HL-60 NB4	EGCG 0–40 µmol/L (24–36 h) IC50 = 21.5 µmol/L (24–36 h) IC50 = 30.5 µmol/L (24–36 h)	Induced apoptosis detected by ↑ of DAPK2 and the level of 67LR expression	[16]
NB4 NB4 R1 NB4 R2 Primary APL/leukemia cells	Catechins 0–400 µM (0–48 h) IC50 < 125	Induced apoptosis through ROS production (↑): mitochondrial dysfunction—loss of Δψm; release cytocromo C; Blc-xL ↓; caspase-3, -8 and -9 ↑; PARP cleavage (↑); morphologic changes—condensed chromatin, nuclei fragmented and apoptotic bodies; and ↓ PML/RARα	[33]
NB4	EGCG 0-40 µM (24 h)	Induced apoptosis through the SHP-1-p38αMAPK-Bax cascade (↑ Bax, SHP-1 (Src homology 1 domain-containing protein tyrosine phosphatase) expression and levels of phosphorylated (p)-p38α MAPK)	[18]
JURKAT	EGCG 0–100 µM (0–72 h) IC50 = 82.8 ± 3.1 µM 24 h 68.8 ± 4.0 µM 48 h 59.7 ± 4.8 µM 72 h	Induced apoptosis through Fas/Fas ligand activation: ↑ Fas expression and caspase-3 ↑	[39]
K562 K652R KCL-22 BaF3/p210 BaF3/p210^T3151^ Primary bone marrow CML cells	EGCG 0–100 µmol/L (0–48 h) IC50 = 62.62 µmol/L K562 91.12 µmol/L K562R 53.76 µmol/L KCL-22 12.80 µmol/L BaF3/p210 29.82 µmol/L BaF3/p210^T3151^	Induced apoptosis by regulating Bcr/Abl (degradation)-mediated JAK2/STAT3/AKT (↓) and p38-MAPK/JNK (↓) signaling pathways. Induced autophagy function (↑ Atg5 and LC3), as well as ↓ MMP (mitochondrial membrane potential), ↑ HSP60 (mitochondrial protein marker), histone H3 and AIF (apoptosis-inducing factor)	[51]
HL60	EGCG 100 µM (5 days) IC50 = 190.4 ± 0.03 µM (5 days)	Induced apoptosis, ↓ AKT and ↑ CASP3, CASPP8, p21 and PTEN gene expression; ↓ ABCB1 and ABCC1 (genes of multi-drug resistance)	[24]
**In vivo—Xenograft and systemics model**			
Murine myeloid leukemia WEHI-3B JCS cells pre incubated with EGGC in Balb/c mice (injected i.p.)	EGCG 40 µM (4 h)	Reduction in the tumorigenicity—↓ the leukemic cell growth	[50]
APL cells (UF-1) in Nod. Scid mice (xenografted)	EGCG 10 mM as the sole drink for 12 days	Reduction in tumor weight and inhibition of cell proliferation. During the treatment, the mice appeared healthy, and there was no change in the tissue organs	[26]
APL cells (NB4) in Nude mice (xenografted)	Catechins 10 mM as the sole drink for 21 days	Reduction in tumor weight and induction of apoptosis During the treatment, the mice appeared healthy, and there was no infiltration in any of the organs	[33]
APL cells (HL60) in NOD. CB17-Prkdcscid/J mice (xenografted)	GT 100 mg/kg as gavage	Reduction in tumor weight and induction of apoptosis (↑ cytochrome c ↓ Bcl-2, ↑ Bax and pJNK, ↑ caspase 3), cell cycle arrest (↓ CDK2 and cyclin A and ↑ p21) and autophagy (↑ LC3-II)	[54]
hCG-PML/RAR transgenic mice cells in NOD. CB17-Prkdcscid/J mice	GT 250 mg/kg/d intraperitoneally, for 5 days	Reduction in spleen weight and induction of apoptosis of blasts in spleen and bone marrow (↑ of caspase-3, -8 and -9) and ↓ CXCR4/HIF-1α pathway in response to ↓ ROS levels	[55]
hCG-PML/RAR transgenic mice cells in NOD. CB17-Prkdcscid/J mice	EGCG 25 mg/kg/d intraperitoneally, for 5 days	Reduction in spleen weight and induction of apoptosis of spleen cells by modulating Bax (↑), Bad (↑), Bcl-2 (↓) and c-Myc (↓)	[7]

## 3. Pro-Oxidant or Antioxidants Effects of Green Tea Catechins

Reactive oxygen species (ROS) play a complex role in cancer development and progression. While low levels of ROS are required for normal cellular processes, excessive ROS production can lead to oxidative stress, which can damage cellular components and promote cancer cell growth and survival. ROS can induce DNA damage, mutations and alterations in the gene expression that can contribute to tumorigenesis. Therefore, ROS levels and their effects on cancer cells depend on a variety of factors, including the type and stage of cancer in addition to the cellular environment [58].

Green tea polyphenols have been suggested as a potential compound for cancer prevention, due to their antioxidant activity and, more recently, pro-oxidant effects [59,60] (Table 2). Among the catechins found in green tea, EGCG is a powerful antioxidant agent related to therapeutic qualities of tea [61].

Initial evidence of EGCG’s antioxidant properties was obtained from experiments showing that this compound could inhibit soybean lipoxygenase, which plays a role in the development of unpleasant flavor in food due to the oxidation of polyunsaturated fatty acids, with an IC50 of 10 to 20 μmol [62]. Subsequent studies revealed that EGCG could inhibit oxidative DNA base modification induced by TPA in HeLa cells, reduce lipid peroxidation induced by tert-butyl hydroperoxide and block the production of reactive oxygen species generated from NADPH-cytochrome P450-mediated oxidation of 2-amino-3-methylimidazo [4,5-f]quinoline [63].

The antioxidative properties of EC, EGC, ECG and EGCG have been demonstrated through various in vitro and chemical assays. The underlying chemistry responsible for this activity primarily involves hydrogen atom transfer or single electron transfer reactions, or a combination of both, which specifically target the hydroxyl groups. These hydroxyl groups are present in the B-rings of EC and EGC and in both the B- and D-rings of ECG and EGCG. Tea catechins, acting as chain-breaking antioxidants, are believed to disrupt harmful oxidation processes primarily through hydrogen atom transfer (HAT) mechanisms. Among these mechanisms, the most significant one is the inhibition of lipid peroxidation [59]. In addition, their polyphenolic structure enables electron delocalization, which can neutralize free radicals. Specifically, EGCG has been observed to decrease various reactive oxygen species such as superoxide radical, singlet oxygen, hydroxyl radical, peroxyl radical, nitric oxide, nitrogen dioxide and peroxynitrite. Additionally, tea polyphenols possess strong chelating properties towards metal ions, thereby inhibiting the formation of reactive oxygen species [64].

On the other hand, the pro-oxidant effects of green tea polyphenols have been further demonstrated more recently. EGCG cytotoxic actions are associated with the autoxidation of the compound as demonstrated previously by Wei et al. in 2016 [65]. This process leads to the production of hydrogen peroxide, as well as the formation of EGCG auto-oxidation products (EAOPs). Chemically, oxygen obtains electrons from EGCG, resulting in superoxide anion formations, which eventually are reduced, resulting in accumulation of hydrogen peroxide, inhibiting thioredoxin reductase, a putative target for cancer prevention [65].

Sang et al. [64] conducted a study in which they investigated the factors affecting the stability of EGCG. According to their findings, the stability of EGCG is influenced by several factors, involving EGCG concentration pH of the system, presence of oxygen and temperature during incubation. For instance, when the concentration of EGCG was increased, the half-life of EGCG in water at 37 °C was increased 5-fold. Similarly, when EGCG (20 μM) was incubated in phosphate buffer (pH 7.4) at 37 °C, the half-life was observed to be 30 min in an air atmosphere, whereas in a nitrogen atmosphere, there was no significant degradation of EGCG for at least 360 min.

Other authors also demonstrated that these polyphenol compounds are unstable and undergo auto-oxidative processes, resulting in the production of reactive oxygen species, playing an important role in the pro-apoptotic effects against cancer cell lines [59] and in leukemic cell lines as previously described.

EGCG treatment induced a slight increase in intracellular reactive oxygen species and GSSG (glutathione disulfide) but without a significant statistical outcome in the U937 cell line. In the same study, EGCG significantly decreased the levels of time-dependent GSH (glutathione—the abundant redox-regulating molecule) until 30 min after treatment, demonstrating an antioxidant intracellular redox state [66].

The induction of oxidative DNA damage was demonstrated in HL60 cells treated with catechins. The study suggests that catechin treatment increased oxidative DNA damage, augmenting the 8-oxodoG content. In addition, catechin could induce metal-dependent H_2_O_2_ in the presence of Cu(II) generation during the redox reactions [67]. EGCG treatment in Jurkat cells has shown that both H_2_O_2_ and Fe(II) mechanisms are involved. This leads to an increase in H_2_O_2_ levels, suggesting that the cytotoxic effect of catechins in specific tumor cells is due to their capacity to generate H_2_O_2_. The subsequent elevated levels of H_2_O_2_ then instigate the formation of highly toxic hydroxyl radicals through an Fe(II)-dependent process. This cascade ultimately induces apoptotic cell death [68].

In 2005, a study demonstrated that EGCG decreased ROS levels in HL-60 cells associated with an increase in the number of apoptotic cells [69].

In an in vitro model using retinoic acid (RA)-resistant acute promyelocytic leukemia—UF-1 cells, EGCG induced apoptosis in association with loss of mitochondrial transmembrane potential and elevated ROS production [26]. The same group demonstrated in a study using myeloma (RPMI8226) and lymphoma cell lines (HS-sultan) that EGCG increased apoptosis associated with loss of mitochondrial membrane potential, suggesting that the effect of the compound was related to an increase in the release of cytochrome c, Smac/DIABLO and AIF from mitochondria into the cytosol. Increased levels of intracellular ROS were further observed, associated with induction of apoptosis mediated by modification of the redox system. In the same study, EGCG additionally increased ROS levels in fresh myeloma cells derived from patients with multiple myeloma, associated with an increase in apoptosis [45]. The same group further investigated the mechanism of ROS-mediated apoptosis and showed that different antioxidants blocked the effect of ROS liberation from EGCG treatment: catalase (H_2_O_2_ scavenger), MnSOD (O_2_ scavenger), taurine (HOCl scavenger), methionine (HOCl scavenger), ABAH (myeloperoxidase inhibitor), desferrioxamine (DFO; iron chelator) and apocynin (NADPH inhibitor) in HL-60 cells, indicating that EGCG produced highly toxic levels in HL60 cells [27].

In human lymphoblastoid B-cells (Ramos cells), EGCG induced dose- and time-dependent apoptotic cell death accompanied by loss of mitochondrial transmembrane potential and release of cytochrome c to the cytosol. The compound also increased intracellular ROS. To further explore the mechanisms of action, an inhibitor of NAD(P)H oxidase and an antioxidant (diphenylene iodonium chloride) partially reversed the effect observed previously, indicating the participation of oxidative stress in the apoptotic response induced by EGCG [70]. Similarly, ECGC presented pro-oxidant effects in HL60 cell, significantly enhancing intracellular hydrogen peroxide [71].

The use of ECG in HL60 cells demonstrated the inhibition of chlorination of hypochlorous acid (HOCL), a strong oxidant derived from myeloperoxidase that can chlorinate DNA bases in an inflammatory state and protect DNA from damage [72].

In IM9 multiple myeloma cells, EGCG reduced the protein levels of peroxiredoxin V (Prdx V, which catalyzes the reduction in hydrogen peroxide), inducing ROS accumulation and cell death [73].

Tofolean et al., 2016 [74], evaluated the effect of EGCG in Jurkat T-cells in mitochondrial dysfunction and oxidative stress. The results demonstrated an early depolarizing effect on mitochondria membrane after an acute exposure to an immediate increase in the NADH level, probably due to inhibition of respiratory complex III.

EGCG is frequently subject to challenges such as susceptibility to oxidation and rapid degradation in aqueous solutions. In this scenario, the effect of EGCG oxides in a T-cell acute lymphoblastic leukemia cell line (HPB-ALL) was demonstrated. An EGCG oxide showed anticancer activity with an IC50 value of around 40 μM, with proliferation inhibition associated with downregulation of Notch1 and Ki67 expression [75]

In an in vivo experimental model of acute promyelocytic leukemia (APL), specifically the PML/RARα model, green tea was found to increase the levels of reactive oxygen species (ROS) within bone marrow Gr1^+^ cells, simultaneously decreasing the number of CD34^+^ and CD117^+^ cells. These findings suggest that green tea has anti-proliferative effects upon leukemia in vivo. This effect is probably achieved by inhibiting the expansion of malignant clones, potentially through modulating the production of ROS within cells [55]. In the same experimental model, EGCG increased ROS levels in bone marrow cells—CD34^+^, CD177^+^ and Gr-1^+^ [7]. In addition, the promyelocytic cell line NB4 treated with EGCG increased ROS levels. When the EGCG treatment was associated with N-acetyl-L-cysteine (an antioxidant compound), ROS levels were inhibited, indicating that EGCG activities involve or are a response to oxidative stress [7]. Table 2 summarizes the current understanding on the effects of green tea catechins on pro- and antioxidant activities.

**Table 2 pharmaceuticals-16-01021-t002:** Pro-oxidant and antioxidant effects of green tea catechins.

Model	Compound/Component	Effect	Reference
**In vitro**			
U937	EGCG 400 µM (5–30 min)	Decreased levels of glutathione	[66]
HL60	Catechins 1 µM (18 h)	Increased oxidative DNA damage augmenting the 8-oxodoG content	[67]
Jurkat	EGCG 12.5–50 µM (6 h)	Increased H_2_O_2_ levels	[68]
HL60	EGCG 50 μM (1 h)	Decreased ROS levels, associated with an increase in apoptotic cells	[69]
UF-1	EGCG 50 μM (3 h)	Increased ROS levels, loss of potential mitochondrial membrane	[26]
HS-Sultan RPMI8226	EGCG 20 or 100 μM (1–4 h)	Increased ROS levels, loss of potential mitochondrial membrane, increased cytochrome c, Smac/DIABLO and AIF	[45]
Fresh myeloma cells from bone marrow from multiple myeloma patients	EGCG 20 μM (8 h)	Increased ROS levels associated with an increase in apoptotic cells	[45]
HL60	EGCG 50 μM (24 h)	ROS production was blocked by antioxidants	[27]
Ramos	EGCG 40 or 80 μM (0.5–4 h)	Increased ROS levels, loss of mitochondrial transmembrane potential, increased release of cytochrome c to the cytosol	[70]
HL60	EGCG 20 μM (1–2 h)	Increased ROS levels	[71]
HL60	ECG 200 μM (2 h)	Inhibited chlorination of hypochlorous acid	[72]
JURKAT	EGCG 150 μM (1 h)	Increased depolarization on mitochondria	[74]
IM9	EGCG 180 μM (9, 12 and 24 h)	Decreased protein levels of peroxiredoxin V	[73]
HPB-ALL	EGCG oxide 5, 10, 20, 40, 60 μM (12 h)	Inhibited proliferation associated with downregulation of Notch1 and Ki67 expression	[75]
NB4	EGCG 12.5–50.0 μg/mL (2 h)	Increased ROS levels	[7]
**In vivo—Xenograft and systemics model**			
hCG-PML/RAR transgenic mice cells in NOD. CB17-Prkdcscid/J mice	GT 250 mg/kg/d intraperitoneally, for 5 days	Increased intracellular ROS in Gr-1^+^ bone marrow cells and decreased intracellular ROS in CD34^+^ and CD117^+^ cells	[55]
hCG-PML/RAR transgenic mice cells in NOD. CB17-Prkdcscid/J mice	EGCG 25 mg/kg/d intraperitoneally, for 5 days	Increased ROS levels in bone marrow cells—CD34^+^, CD117^+^ and GR-1^+^	[7]

## 4. Epigenetics

Epigenetic processes manage heritable gene expression patterns that are not determined by DNA sequence variations. These processes enable genetically identical cells to display unique gene expression profiles, which dictate cellular specialization. Epigenetic marks, including DNA methylation, histone post-translational modifications and nucleosome positioning, serve as reversible storage and transmitters of inheritable data. Enzymes responsible for creating these epigenetic marks facilitate DNA methylation or histone post-translational modifications such as methylation or acetylation of N-terminal histone “tails” that project from the nucleosome structure. Epigenetic marks can be identified by protein assemblies that modify chromatin structure or control enzymatic activities such as transcription factor attachment and RNA polymerase effectiveness. Enzymes such as histone deacetylases or demethylases can eliminate these epigenetic marks. Furthermore, chromatin reorganizers can shift or swap histones, contributing to the dynamic nature of the epigenetic landscape [76]. Since epigenetic marks can be reversed, numerous anti-cancer approaches aimed at restoring the epigenome to a healthier state are in development and undergoing clinical testing.

### 4.1. DNA Methylation and Gene Regulation

DNA methylation primarily takes place at the 5′-position on the pyrimidine ring of cytosine residues in the context of symmetric CpG dinucleotide (CG) sequences, where a methyl group is added to create 5-methylcytosines. CpG dinucleotides frequently accumulate in CpG-dense DNA “islands” situated near the transcription start sites (TSSs) [77]. An increase in methylation of CpG islands can potentially suppress the expression of adjacent genes by obstructing transcription factor attachment and attracting methyl-CpG binding proteins. These proteins consequently influence the activity of histone-modifying enzymes associated with repression [78]. This reaction is facilitated by a group of enzymes known as DNA methyltransferases (DNMTs), which utilize S-adenosyl-methionine as the source of the methyl group. These enzymes are responsible for creating and preserving DNA methylation patterns. DNMT3A and DNMT3B initiate new DNA methylation patterns, while DNMT1 maintains them by identifying DNA methylation and guiding the methylation of the daughter strand after replication [79]. The TET enzymes facilitate DNA demethylation by transforming methyl cytosine into hydroxymethylcytosine, a base modification that DNMT1 does not recognize [80].

Leukemia, as with any other cancer, displays aberrant methylation profiles. For instance, hypermethylation of hypermethylated in cancer 1 (HIC1) has been noted in AML patients in advanced stages who relapsed after undergoing chemotherapy and achieving complete remission [81]. The hypermethylation of the promoter region of homeobox (HOX) A4 gene is associated with a negative prognosis, while the WIT1 gene has been associated with chemoresistance in AML [82,83]. Additionally, the methylation of cyclin-dependent kinase inhibition (CDKN)2B (p15) displays a high degree of heterogeneity among AML patients [83,84]. The methylation of CDKN2B (p15) may be involved in the progression of the disease and leukemogenesis [85]. Numerous studies have highlighted the correlation between the transcriptional inactivation of various TSGs and the extensive de novo methylation of their promoter regions. Intriguingly, AML and CML patients have shown an increase in the expression of DNMT1, DNMT3a and DNMT3b [86]. Notably, AML cells with a hypermethylated CDKN2B (p15) gene exhibited higher levels of DNMT1 and DNMT3B expression. These findings suggest that the upregulation of DNMTs could contribute to CDKN2B methylation.

### 4.2. Covalent Histone Modifications

The proper regulation of the chromatin structure, transcription and DNA repair depends on the control of lysine acetylation in the histone tail region. This highly dynamic modification is governed by histone lysine acetyltransferases (HATs) and histone deacetylases (HDACs). HATs transfer the acetyl group from acetyl-coenzyme A to the amino group of a lysine in the histone, thereby neutralizing the lysine’s positive charge and loosening the interaction of histone with DNA. Generally, histone acetylation promotes a relaxed and accessible chromatin structure that is conducive to the binding of proteins such as transcription factors. As a result, chromatin acetylation is usually linked to transcriptional activation, whereas deacetylation is associated with gene repression [87]. HDACs have a significant role in gene expression by eliminating the activating histone acetylation and may also have other cellular functions by regulating acetylation of non-histone and non-nuclear proteins. In cancer, HDACs are often found to be overexpressed and may silence tumor suppressor genes. Furthermore, overexpressed transcription factors or chimeric transcription factors formed by aberrant chromosomal fusions may abnormally recruit HDACs to target genes [87]. The enzymatic activities of histone acetyltransferases and HDACs have an impact on various gene classes that are associated with crucial biological processes such as development, proliferation and differentiation. Any imbalance or dysfunction in these enzymatic activities could be held responsible for cancer-related events.

Certain forms of leukemia have been linked to the activities of histone deacetylases. The PML–RAR fusion protein leads to leukemogenesis by disrupting the normal epigenetic regulation of retinoic acid (RA) target genes. The presence of PML in the chimera protein leads to oligomerization of RAR, leading to an abnormal interaction with the NCoR/SMRT corepressive complex that recruits HDACs. In cells with PML–RAR, physiological amounts of RA are unable to dissociate the chimera protein from NCoR/SMRT to release the repressive complex. This sustained recruitment of HDACs causes an aberrant repression of RA target genes, leading to blocked hematopoietic differentiation and, ultimately, malignancy. As a result, treatment with all-trans retinoic acid has been successfully introduced for APL patients with the PML–RAR fusion protein, leading to the differentiation of blasts into granulocytes and often inducing patient remission [87,88]. The aberrant recruitment of HDACs to target genes has been suggested to be the result of oligomerization of chimeric transcription factors, which could be a mechanism underlying a broader range of AMLs. For instance, the non-APL leukemia fusion protein AML1–ETO has been shown to act with a similar abnormal HDAC-recruiting mechanism, leading to blocked hematopoietic differentiation [89]. Furthermore, the combination of RA with HDAC inhibitors in patients with RA-resistant APL (such as those bearing the PLZF–RAR chimera) has produced encouraging clinical responses, although some blasts remain resistant to differentiation [88,90].

The abnormal activation or overexpression of histone-modifying enzymes results in the dysregulation of gene transcription, which may involve the suppression of TSGs and activation of developmental genes that promote cancer progression. Targeting histone-modifying enzymes with small inhibitory molecules represents a promising approach for the treatment of various types of cancers, including leukemia [91,92,93]. This strategy aims to reverse the abnormal conditions caused by these enzymes, offering a potential therapeutic solution for cancer patients.

From a medical perspective, epigenetics therapy presents a highly appealing and promising path as, either individually or in conjunction with traditional anticancer drugs, it could represent a substantial improvement over conventional anticancer treatments, which are typically highly toxic [94]. Both epidemiological and experimental data have convincingly demonstrated that dietary factors and plant-based polyphenols derived from the diet possess strong anticancer properties. Some of these effects are mediated through the regulation of epigenetic mechanisms within cancer cells [95].

EGCG, the primary polyphenol in green tea, has been extensively researched as a potential demethylating agent. EGCG is methylated by catechol-O-methyltransferase (COMT), the enzyme responsible for inactivating catechol molecules, such as dietary polyphenols. This enzyme adds a methyl group to the catecholamine group, which is donated by S-adenosyl methionine (SAM). SAM’s demethylation leads to the formation of S-adenosyl-L-homocysteine (SAH), which is a potent inhibitor of DNMT. The production of SAH has been suggested as one of the mechanisms for the demethylating properties of this compound. Additionally, EGCG can create hydrogen bonds with various residues in the catalytic pocket of DNMT, thereby acting as a direct inhibitor of DNMT1 [96,97]. Inhibiting DNMT may prevent the methylation of the newly created DNA strand, leading to the reversal of hypermethylation and the re-expression of silenced genes. Moreover, EGCG has been demonstrated to be an effective inhibitor of human dihydrofolate reductase. Similar to other antifolate compounds, EGCG interacts with folic acid metabolism in cells, inhibiting DNA and RNA synthesis and altering DNA methylation [98]. In addition to their ability to induce changes in DNA methylation, evidence indicates that EGCG can further regulate gene expression through changes in histone modifications [99,100,101,102]. The text below and Table 3 summarize the current understanding on the effects of EGCG on histone modifications and DNA methylation, which may play a significant role in the potential use of this compound.

EGCG treatment has been demonstrated to decrease DNMT1 levels, as well as euchromatic histone lysine methyltransferase 2 (G9a), HDAC1/2 and polycomb repressive complex 2 (PRC2) in acute promyelocytic leukemia cells. This results in the attachment of acetylated H3K14 histone and hyperacetylated H4 to the promoter region of chromatin assembly factor (CAF), CCAAT/enhancer binding protein, α/ε (C/EBPα/ε) and p27 [103]. Another study with the U937 cell line showed that EGCG treatment decreased methylation of the SOCS1 gene promoter and increased SOCS1 expression in a time-dependent manner [104]. EGCG treatment also revealed positive epigenetic modulation of the epigenetic players DNMT1, HP1α, H3K9me3, EZH2 and SUZ12, on the NB4 cell line. However, these epigenetic changes were not observed in K56a cells [105]. Moreover, 25 μM of EGCG prompted differentiation of leukemic cells towards a granulocytic pattern, comparable to the effects of 1 μM ATRA. Additionally, EGCG downregulated the expression of the clinical marker PML/RARα in NB4 cells and diminished HDAC1 expression in leukemic cells [106].

The use of EGCG as an epigenetic therapy in leukemia has emerged as a promising approach with substantial potential. EGCG possesses remarkable epigenetic modulatory properties, specifically targeting key epigenetic enzymes and regulatory pathways. Additionally, the combination of EGCG with conventional chemotherapeutic agents may have synergistic effects, enhancing therapeutic efficacy while potentially reducing side effects.

Further research is needed to validate and optimize the application of EGCG as an epigenetic therapy for leukemia. Future studies should focus on elucidating the precise molecular mechanisms of action, identifying optimal dosages and treatment regimens and evaluating the long-term safety and effectiveness of EGCG.

## 5. Differentiation of Acute Promyelocytic Leukemia by Green Tea Catechins

Induction of differentiation of leukemic cells may represent one of the most promising approaches for the treatment of acute promyelocytic leukemia, as demonstrated with the use of all-*trans* retinoic acid (ATRA). Interestingly, green tea polyphenols promote differentiation (Table 4).

EGCG treatment increases surface expression of myeloid differentiation markers such as CD11b, CD14, CD15 and CD66b in HL60 and NB4 cell lines [7,16] and induces PML/RARα oncoprotein degradation independent of apoptosis [33]. Wei et al. (2015) demonstrated that inhibition of Pin by EGCG causes PML/RARα degradation in cell lines [107]. Yao et al. (2017) also observed that the degradation of PML/RARα restored the PML and PTEN function supporting the differentiation [108]. In addition, Moradzadeh et al. (2017) reported that EGCG induces differentiation by HDAC1 inhibition, which is recruited by the PML/RARα fusion and suppresses genes involved in differentiation [106].

The EGCG and ATRA combination intensifies surface expression of myeloid differentiation markers, such as CD11b [16,108] and CD15 [7,16]; expression levels of key genes involved in regulation of terminal granulocyte differentiation, such as CEBPE [16,108], CSF3R [16] as well as genes that may be necessary/required for differentiation, such as PTEN [108] and DAPK2 [16].

In an animal model of acute promyelocytic leukemia (APL) the EGCG treatment causes PML/RARα degradation in bone marrow cells [7,107]. Not only does the treatment reduce immature cells and undifferentiated promyelocytes in the bone marrow, but it further increases mature myeloid cells in response to a reduction in PIN1 and substrate oncoproteins for PIN1, such as cyclin D, NFκB, c-Myc and AKT. Moreover, EGCG treatment increases ROS [7].

## 6. Epidemiological Data and Clinical Trials with Green Tea

The vast majority of epidemiological studies (case–control studies or meta-analyses) describe that tea consumption has a protective function in reducing the risk of myelodysplastic syndromes (MDSs) [109,110] or leukemia in different populations [110,111,112,113,114,115,116,117,118,119,120,121,122] (Table 5) despite the mechanism not having been completely understood yet.

In relation to clinical trials, a study in ten elderly acute myeloid leukemia patients with myelodysplasia-related changes (AML-MRCs) using green tea extract associated or not with low-dose of cytarabine (for at least 6 months) showed an increase in CD8^+^ T-cells, natural killer cells and classical monocytes and a reduction in Treg cells and mRNA expression of TGF-beta and IL4 in bone marrow and/or peripheral blood [123].

For lymphoid leukemia, Shanafelt et al. (2006) became aware that four patients with low-grade B-cells malignancies had begun, by their own initiative, oral ingestion of products containing tea polyphenols and presented an objective clinical response [124]. Clinical phase I and II trials of daily oral EGCG (Polyphenon E—6 months) in patients with asymptomatic Rai stage 0 to II CLL showed a decline in the absolute number of lymphocytes and the lymphadenopathy [125,126]. D’Arena et al. (2013) additionally reported that green tea extract, orally for 6 months, reduced the absolute number of B-lymphocytes and circulating Treg cells accompanied by a decline of IL-10 and TGF-beta serum levels in a cohort of patients with low-risk CLL (Rai stage 0, *n* = 12/10 per group (*n* = 32 and 42 patients respectively) [127]. Lemanne et al. (2015) reported a case of a 48-year-old man with CLL who presented a complete clinical and molecular regression, 20 years after a diagnosis, without conventional therapy and using EGCG [128]. These studies suggest that green tea catechin extracts are safe/tolerated (in the tested doses) in healthy individuals [129,130] and leukemia patients [123,125,126,128], as well as inexpensive and effective.

**Table 5 pharmaceuticals-16-01021-t005:** Epidemiological data and clinical trials with green tea.

Disease	Compound/Component	Effect	Reference
Prevention			
AML Hospital-based case–control study (*n* = 111 cases and 439 controls)	Tea	Regular and high daily intake of tea reduced the risk of adult AML among males and females in New York	[112]
Leukemia Hospital-based case–control study (*n* = 107 cases and 110 orthopedic controls)	Green tea	The frequency, longer duration and higher quantity of green tea intake reduced the risk of ALL and CML/CLL in Southeast China	[119]
Leukemia Hospital-based matched case–control study (*n* = 107 cases and 110 inpatient controls)	Green tea	High consumption of green tea reduced the risk of adult leukemia	[120]
Leukemia Population-based case–control study (*n* = 252 cases and 637 controls)	Tea	The highest intake of tea, especially green tea, reduces the risk of leukemia in Southwestern Taiwan	[121]
AML NIH–AARP cohort (*n* = 338 cases and *n* = 491,163 people)	Tea	No association with consumption of tea and risk of leukemia in US	[115]
Childhood acute leukemia Population-based case–control study (*n* = 190 cases and 842 controls)	Tea	No association with consumption of tea and risk of leukemia in Southern Taiwan	[113]
Hematologic malignancies Population-based case–control study (*n* = 41,761)	Green tea	Green tea consumption reduces the risk of hematologic malignancies—lymphoid and myeloid neoplasms—in Japan	[116]
Leukemia Meta-analysis	Tea	High tea consumption reduces the risk of leukemia, indicating a protective role of tea against leukemia	[122]
De novo MDS Hospital-based case–control (*n* = 208 cases and 208 controls)	Tea	Regular and high intake of tea reduces the risk of MDS in China	[109]
Adult leukemia Hospital-based case–control (multicenter) (*n* = 442 cases and 442 outpatient controls)	Green tea	The regular daily intake of green tea reduces the risk of leukemia regardless of GSTM1 and GSTP1 polymorphic status in China	[[114]
AML and MDS Population-based cohort study (*n* = 95,807)	Green tea	No association between green tea consumption and the risk of MDS and AML in Japan	[110]
Adult leukemia Population-based cohort study (*n* = 651 cases and 1771 controls)	Tea	A protective effect of tea intake on the risk of AML in Italy	[117]
Hematologic neoplasm Community-based prospective study (*n* = 110,585 individuals)	Green tea	A protective effect of tea intake against hematologic neoplasm, specifically of AML and follicular lymphomas in Japan	[118]
Cancers Meta-analysis of observational studies.	Tea	The tea consumption was associated with a lower risk of cancer, like leukemia, showing a protective effect	[111]
**Clinical trials**			
Patients with low-grade B-cells malignancies (*n* = 4)	Oral ingestion of products containing tea polyphenols by their own initiative	An objective clinical response	[124]
Asymptomatic Rai stage 0 to II CLL (phase I trial) (*n* = 33)	Polyphenon E—6 months 2000 mg twice per day	Reduction in absolute number of lymphocytes and lymphadenopathy in most patients	[125]
Asymptomatic Rai stage 0 to II CLL (phase II trial) (*n* = 42)	Polyphenon E—6 months 2000 mg twice per day	Reduction in absolute number of lymphocytes and lymphadenopathy in most patients	[126]
Patients with Rai stage 0 CLL (*n* = 12 per group)	Green tea extract—6 months four capsules/day for the first months and six capsules/day for the following 5 months (400 mg of green tea total concentrate per capsule)	Reduction in the absolute number of B-lymphocytes, circulating Treg cells and IL-10 and TGF-β serum levels	[127]
48-year-old man with CLL (*n* = 1)	1200 mg/day of EGCG	Patient achieved a complete clinical and molecular regression, 20 years after a diagnosis, without conventional therapy and using EGCG	[128]
Elderly acute myeloid leukemia patients with myelodysplasia-related changes (AML-MRCs) (*n* = 10)	Green tea extract—6 months (1000 mg/day—4 capsules/day)	Reduction in the immunosuppressive profile by ↓ Treg cells, CXCR4^+^ Treg cells and mRNA expression of TGF-β and IL4, and activation of cytotoxic phenotype by ↑ CD8^+^ T-cells, natural killer cells and classical monocytes in bone marrow and/or peripheral blood	[123]

## 7. Conclusions

Green tea catechins, particularly epigallocatechin-3-gallate (EGCG), exhibit a range of beneficial effects that position them as a promising candidate for the prevention and treatment of leukemia and myeloid dysplastic syndrome (MDS). Extensive research has demonstrated that green tea catechins induce apoptosis by increasing pro-apoptotic proteins, depolarizing the mitochondrial membrane and inducing reactive oxygen species (ROS) generation. This apoptotic effect provides a means to selectively eliminate leukemia cells, while sparing normal healthy cells. Furthermore, the redox signaling properties of green tea catechins contribute to their potential in promoting leukemia cell differentiation. By modulating the redox balance and enhancing antioxidant defenses, these compounds create an environment conducive to the restoration of normal cellular maturation. This differentiation-promoting effect can help counteract the uncontrolled proliferation and impaired differentiation characteristic of leukemia and MDS.

Green tea catechins have also shown promise in reversing epigenetic alterations, including DNA methylation and histone modifications. These epigenetic changes play a pivotal role in the dysregulation of gene expression patterns observed in leukemia and MDS. By modulating epigenetic marks, green tea catechins offer a unique approach to reprogramming aberrant gene expression and potentially restoring normal cellular function. In addition, clinical trials investigating the effects of green tea catechins in leukemia and MDS patients have demonstrated promising results. These trials revealed the protective effect of green tea catechins against leukemia development, as well as their potential as an adjunctive therapy in combination with conventional treatments. The ability of green tea catechins to selectively target leukemia cells, promote differentiation and modulate epigenetic alterations has significant implications for the prevention and treatment of these hematological malignancies.

In conclusion, green tea catechins, particularly EGCG, possess a multifaceted range of actions that make them attractive candidates for the prevention and treatment of leukemia and MDS. Their ability to induce apoptosis, promote differentiation, modulate redox signaling and reverse epigenetic alterations underscores their potential therapeutic value. Additional investigation is crucial to enhancing the bioavailability of catechins and comprehending the variables affecting individual responses, and it is also essential to consider the potential risks associated with high-dose catechin intake and to approach their use in a balanced, informed manner. Further in-depth studies and clinical trials are needed to comprehensively investigate the effectiveness, safety and appropriate dosage strategies of green tea catechins in the prevention and treatment of leukemia and myelodysplastic syndromes (MDSs).

## Figures and Tables

**Figure 1 pharmaceuticals-16-01021-f001:**
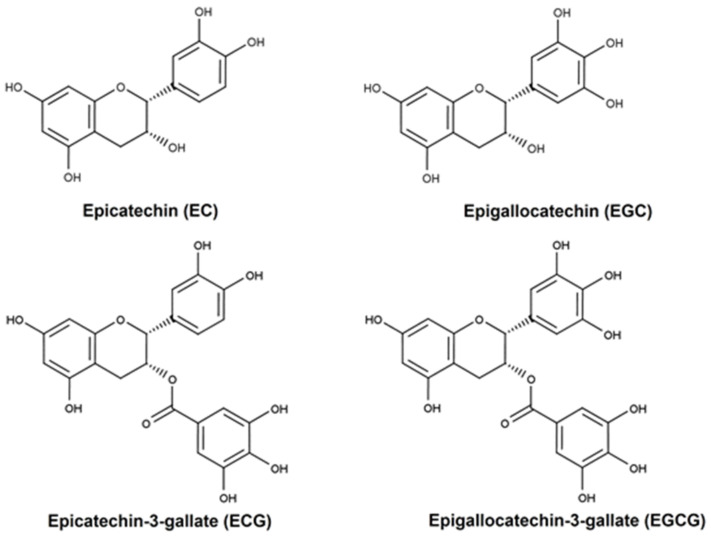
Chemical structures of epicatechin (EC), epigallocatechin (EGC), epicatechin-3-gallate (ECG) and epigallocatechin-3-gallate (EGCG). The green tea catechin structures are characterized by the dihydroxyl or trihydroxyl substitution on the B-ring and the m-5,7-dihydroxil substitutions on the A ring [5]. Created with ChemDraw^®^.

**Table 3 pharmaceuticals-16-01021-t003:** Effects of green tea catechins in epigenetics.

Model	Compound/Component	Effect	Reference
**In vitro**			
NB4 HL60	EGCG 30–40 µM (6–96 h)	↓ DNMT1, HDAC1, HDAC 2, G9a, H3K9me^2^, PCR2 and ↑ H4, H3K14 (p27, PCAF, c/EBPalfa, c/EBPe) and (EZH, SUZ12, EED)	[103]
U937	EGCG 25 µM (12–72 h)	↓ Promoter methylation SOCS1	[104]
NB4 K562	EGCG 30–40 µM (0–3 days)	↓ DNMT1, H3K9me^3^ HP1alfa, EZH2, SUZ12 only NB4 cell line, no effect on K562	[105]
NB4 HL60	EGCG 0–100 µM (24–72 h)	↓ HDAC1	[106]

**Table 4 pharmaceuticals-16-01021-t004:** Induction of differentiation by green tea catechins.

Model	Compound/Component	Effect	Reference
**In vitro**			
HL60 NB4	EGCG 0–40 µmol/L (24–36h) ATRA—1 µmol/L	Induction of neutrophil differentiation—↑ CD11b surface expression EGCG + ATRA—↑ CD11b and CD15 expression; ↑ CEBPE, CSF3R and DAPK2; and ↓ of undifferentiated promyelocytes and myelocytes	[16]
NB4 NB4 R1 NB4 R2	Catechins 100–200 µM (12–24 h)	Induction of PML/RARα degradation	[33]
NB4	EGCG 0–30 µM (72 h) ATRA—10 µM	Induction of PML-RARα degradation by inhibition of Pin1 (↓)	[107]
HL-60 NB4	EGCG 0–100 µM (24–72 h) ATRA—10 µM	Induction of granulocytic maturation—morphological changes and ↑ NBT reduction ability; and degradation of PML/RARα (↓) and HDAC1 (↓)	[106]
HL60 NB4 THP-1	EGCG 5–40 µM (48 h) ATRA 1 µM/mL	Induction of PML/RARα degradation (↓) and restored PML (↑) and PTEN (↑) function EGCG + ATRA—↑ PTEN, CD11b, CEBPE	[108]
NB4	EGCG 12.5–20 µg/mL ATRA—1 µM	Induction of neutrophil differentiation—↑ of CD11b, CD14, CD15 and CD66 surface expression EGCG + ATRA—↑ CD15 expression	[7]
**In vivo—Xenograft and systemics model**			
CTSG-PML-RARA transgenic mice cells in C57BL/6J mice	EGCG 12.50 mg/kg/d intraperitoneally for 21 days ATRA—5 mg	Induction of PML-RARα degradation in bone marrow cells and ↓ of spleen weight	[107]
hCG-PML/RAR transgenic mice cells in NOD. CB17-Prkdcscid/J mice	EGCG 25 mg/kg/d intraperitoneally, for 5 days	Induction of differentiation—↓ immature cells and undifferentiated promyelocytes in the bone marrow, ↑ mature myeloid cells, ↓ PIN1 and its substrates—cyclin D, NFκB, c-Myc and AKT and ↑ ROS; ↓ of spleen weight	[7]

## Data Availability

Not applicable.

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
