# Peer review of "The Effects of Green Tea Catechins in Hematological Malignancies"

_pharmaceuticals, 2023, doi:10.3390/ph16071021_

Round 1

Reviewer 1 Report

The authors report the cytotoxic effects of polyphenols precisely catechins from green tea on various cancer cell lines. This review article is comprehensive and properly written and it is very relevant. It depicts the potential of green tea as an anticancer nutraceutical.

Some little recommendations

1.       Graphical figure should be improved and provided as a figure within the paper.

2.       The last table before references is wrongly placed

3.       The reference style should respect the journal recommendations and numbers should be used within the text while they appear in the chronological order in the reference list.

4.       IC50 should not be in italics

5.       Recommended paper: "HPLC-DAD phenolic profiles, antibiofilm, anti-quorum sensing and enzyme inhibitory potentials of Camellia sinensis (L.) O. Kuntze and Curcuma longa L." LWT 133 (2020): 110150. https://doi.org/10.1016/j.lwt.2020.110150

Author Response

We greatly appreciate the time and effort you have put into reviewing our manuscript titled " The effects of green tea catechins in hematological malignancies ". We are grateful for your constructive feedback and we have taken all your suggestions into account to improve the quality of our work.

  1. Graphical figure: We understand your concerns about the graphical figure in the manuscript. We have revised the figure to provide more detail and clarity and have incorporated it directly into the body of the text to provide a more seamless reading experience. We believe the improved figure will better assist readers in understanding our review and its findings.
  2. Table placement: We apologize for the misplacement of the final table. This has been corrected and the table is now accurately placed within the manuscript.
  3. Reference style: We thank you for bringing this to our attention. We have now thoroughly reviewed and updated all references in accordance with the journal’s guidelines and ensured that they appear in numerical order as they are cited in the text.
  4. IC50 formatting: We apologize for the oversight and have now corrected the formatting error. IC50 values in the text are no longer italicized.
  5. Recommended paper: We appreciate your suggestion to include the recommended paper in our review. We have read the paper and found it relevant. We have therefore included this reference in the appropriate sections of our manuscript and cited it accordingly.

      Again, we would like to express our gratitude for your  recommendations. We believe that the revised version is now ready for publication and will be of interest to the readers of the journal.

Thank you for considering our work.

Reviewer 2 Report

In summary, this article on the effects of green tea catechins in the prevention and treatment of leukemia and myelodysplastic syndrome (MDS) has several strengths. It provides a comprehensive review of the existing literature, addressing the mechanisms of action of green tea catechins, such as apoptosis induction, promotion of cellular differentiation, and modulation of epigenetic alterations. The article also discusses epidemiological studies and clinical trials that support the therapeutic potential of these compounds. Furthermore, the writing is clear and organized, facilitating readers' understanding. However, there may be some minor improvements needed, such as explicitly connecting the different effects of catechins and providing a more in-depth discussion of the limitations and challenges associated with the use of these compounds. Overall, it is a high-quality article that provides a solid foundation for future research and clinical development.

Minor editing of English language required

Author Response

Thank you very much for your comments on our manuscript titled " The effects of green tea catechins in hematological malignancies ". We highly value your constructive feedback and have thoroughly revised our manuscript to address your suggestions.

We have expanded our discussion on the limitations and challenges associated with the use of catechins. We understand that every treatment has potential limitations and it's vital to acknowledge these for a balanced view and to guide future research.

The paragraph included: “Additional investigation is crucial to enhance the bioavailability of catechins and comprehend the variables affecting individual responses, is also essential to consider the potential risks associated with high dose catechin intake and to approach their use in a balanced, informed manner. Further in-depth studies and clinical trials are needed to comprehensively investigate the effectiveness, safety, and appropriate dosage strategies of green tea catechins in the prevention and treatment of leukemia and myelodysplastic syndromes (MDS) “

Again, we express our gratitude and we hope that the revised version of our manuscript will meet with your approval and look forward to your positive response.
